

# Art design integrating visual relation and affective semantics based on Convolutional Block Attention Mechanism-generative adversarial network model

Jiadong Shen and Jian Wang

School of Design and Art, Changsha University of Science and Technology, Changsha, Hunan, China

## ABSTRACT

Scene-based image semantic extraction and its precise sentiment expression significantly enhance artistic design. To address the incongruity between image features and sentiment features caused by non-bilinear pooling, this study introduces a generative adversarial network (GAN) model that integrates visual relationships with sentiment semantics. The GAN-based regularizer is utilized during training to incorporate target information derived from the contextual information into the process. This regularization mechanism imposes stronger penalties for inaccuracies in subject-object type predictions and integrates a sentiment corpus to generate more human-like descriptive statements. The capsule network is employed to reconstruct sentences and predict probabilities in the discriminator. To preserve crucial focal points in feature extraction, the Convolutional Block Attention Mechanism (CBAM) is introduced. Furthermore, two bidirectional long short-term memory (LSTM) modules are used to model both target and relational contexts, thereby refining target labels and inter-target relationships. Experimental results highlight the model's superiority over comparative models in terms of accuracy, BiLingual Evaluation Understudy (BLEU) score, and text preservation rate. The proposed model achieves an accuracy of 95.40% and the highest BLEU score of 16.79, effectively capturing both the label content and the emotional nuances within the image.

Corresponding author
Jian Wang,
paopaochahu@stu.csust.edu.cn

## INTRODUCTION

Thanks to the extensive digitization of a vast array of historical human visual art-works, computer technologies, notably artificial intelligence, particularly deep learning, have been boldly and remarkably employed within the realm of art in recent years (*Zhao et al., 2023*; *Kolesnyk et al., 2022*). On one hand, research pertaining to art analysis has been able to leverage extensive collections of digitized images and image metadata, introducing data-driven principles encompassing mathematics, statistics, data mining, and visualization techniques into the domain of art analysis. On the other hand, data-driven computer technology has also inscribed a magnificent chapter within the realm of art creation, with computers assuming the role of creative agents (*Wang, SB & Lim, 2020*).

In the realm of art history, digital technology has revolutionized the way scholars conduct research and analyze artworks. One of the key advancements in this field is the construction of bespoke art databases that are tailored to the specific research inquiries within specialized domains. These databases are designed to cater to the distinct requirements of scholars, providing them with a wealth of information and tools to enhance their understanding and analysis of artworks.

The utilization of database technology enables art historians to engage in quantitative analysis of data. By quantitatively calculating and organizing data within the databases, scholars can cluster diverse attributes of artworks. These attributes encompass various aspects such as the physical characteristics of the artwork, including materials used and dimensions, the brushwork genres employed by the artists, as well as the temporal and spatial attributes of the artworks. Furthermore, the content depicted within artwork images can also be analyzed and categorized within the databases (*Powell, Gelich & Ras, 2021*; *Kim & Kang, 2022*). This quantitative analytical approach has emerged as a novel research paradigm in the field of digital art history. It goes beyond the traditional methods of art analysis and introduces a more systematic and data-driven approach. The construction of comprehensive art databases not only involves the aggregation of digital images but also encompasses image metadata, previous research findings, artists' biographies, and genres. Moreover, these databases capture the intricate interplay and inheritance relationships between artists, shedding light on the influences, collaborations, and artistic lineages that shape the development of art over time.

The enhanced visual comprehension and reasoning prowess facilitated by these digital art databases offer a wealth of design elements for further reference (*Hudson & Manning, 2019*). Scholars can delve into the intricate details of artworks, examining the relationships between objects within a given scene and unraveling the symbolism and meaning behind the artistic choices. This comprehensive analysis not only benefits the scholars themselves but also contributes to the broader field of art history by uncovering new insights, challenging existing interpretations, and paving the way for innovative research directions.

In conclusion, the utilization of digital technology and the construction of bespoke art databases have transformed the field of art history. These databases enable scholars to engage in quantitative analysis, clustering diverse attributes of artworks, and providing a comprehensive understanding of the artistic landscape.

A scene graph constitutes a graphical data structure responsible for delineating the interconnections among objects within an image. Specifically, this structure employs nodes to represent the objects present in the scene, while the edges symbolize the relationships existing between diverse entities within the graph framework (*Chang et al., 2021*). Scene graphs facilitate the acquisition of comprehensive insights concerning a given image's context. Furthermore, incorporating techniques of image sentiment semantic description to enable computer systems in discerning emotional content within an image and articulating such information through textual means becomes imperative when considering the requirements of artistic analysis within a digital humanistic milieu. It enables a profound exploration of the emotional resonance shared between the designer and the viewer. The art-work titled "Portrait of Edmund Bellamy", crafted by the talented Robbie, stands as a

remarkable manifestation achieved through the utilization of GAN. Robbie, a prodigious AI savant, pushes the boundaries of neural networks and the conventional art realm (*Wu, Seokin & Zhang, 2021*). Barratt, the mastermind behind this feat, furnished the GAN with an extensive assortment of nude portraits and artistic creations sourced from diverse historical eras. This vast repository served as the foundation for training the GAN's neural network, thereby engendering exquisitely surreal depictions of nude portraits and landscape paintings. In anticipation of New York Fashion Week in September 2018, Barratt meticulously compiled a myriad of visual materials, encompassing the House of Paris' designs procured from brochures, advertising campaigns, runway exhibitions, and online catalogs. These meticulously curated resources constituted the training data for the pix2pix neural network, culminating in the birth of an entirely new collection of AI-generated artworks—the Parisienne collection (*Dennis, 2019*).

In the domains of computer vision and sentiment analysis, feature extraction plays a crucial role. Images are typically processed through convolutional neural networks (CNNs) to extract visual features, while sentiment features are derived from textual data using sentiment analysis techniques. However, nonlinear pooling operations, such as max pooling or average pooling, introduce complex transformations during feature extraction that can lead to discrepancies between image features and sentiment features. This inconsistency arises from several key issues. Firstly, nonlinear pooling can distort feature maps by compressing the information, which may result in the loss of important details. Consequently, this information loss can misalign visual features with emotional semantics. Secondly, pooling operations may lead to the loss of crucial details that are essential for accurately capturing the emotional state conveyed by the image. Lastly, when combining image and sentiment features for multimodal analysis, such inconsistencies can impair the effectiveness of the final model, potentially affecting tasks such as emotion-driven image classification.

Scene graph generation models employing the bottom-up methodology have emerged as a prominent research direction in the realm of scene graph generation (*Liu et al., 2021*; *Guo et al., 2021*). This approach enhances the metrics associated with target detection within scene graph generation by initially pre-training target detection using Faster-R-CNN. This preliminary step ensures the precision and accuracy of target detection. Subsequently, a relationship learning model is employed to acquire knowledge and rationalize the relationships between pairs of targets in the image, leveraging statistical insights. Within the bottom-up paradigm, contextual learning of targets and relationships in images has gained considerable popularity. The prevailing context-based approach incorporates local context, encompassing image features corresponding to the targets. However, this approach tends to reason based on statistical relationships, thus downplaying the influence of crucial image features on the inference capabilities of the final scene graph generator. In this article, we propose a novel approach, which takes into account both low-level and high-level semantic features. This integration allows the extracted features to effectively represent both intricate image details and higher-level semantic information. Additionally, we train the model using feature matching techniques to enhance the performance of both the generator and discriminator.

## RELATED WORKS

Following the remarkable achievements of AlexNet in the ImageNet Large Scale Visual Recognition Challenge (ILSVRC) challenge, deep artificial neural networks have demonstrated unparalleled accuracy across a multitude of tasks, owing to their remarkable learning capabilities and their aptitude for generating robust representations of input data (*Manessi & Rozza, 2018*). These deep artificial neural networks have emerged as a cornerstone in diverse domains of computer vision, including image recognition, object recognition, and image processing. Their utilization facilitates the creation of artistic designs characterized by visually diverse renderings, imbued with intuitive representations and profound insights surpassing the limitations of textual interpretations.

With the advancement of artificial intelligence, the prowess of deep learning has become increasingly evident, propelling image description into an entirely new era. *Raju et al. (2019)* introduced the Google Neural Image Caption model, which leverages Google Net for encoding and employs the long short-term memory (LSTM) model for decoding, yielding impressive outcomes. This encoding-decoding framework has since served as a foundation for image semantic description. Subsequently, the attention mechanism gained prominence and found widespread adoption in neural networks (*Manieniyan, Senthilkumar & Sukumar, 2021*). However, the encoding-decoding framework is susceptible to issues such as gradient explosion and gradient disappearance (*Yang et al., 2020*; *Jin, Hu & Zhang, 2020*). Furthermore, the prevailing method for generating utterances in the decoding stage relies on maximum likelihood estimation, wherein each subsequent word is contingent on the previous word. This approach is susceptible to biases, and if a bias emerges in one word, it can accumulate and adversely impact subsequent words. Consequently, the quality of the generated output deteriorates over time. To address these concerns, the application of GAN for image semantic description has proven effective. *Li, Jang & Sung (2019)* introduced the conditional GAN model, which facilitates the generation of multiple descriptions for images by controlling the variance of the hidden layer vector in the initialized generator LSTM. Similarly, *Ma et al. (2022)* incorporated conditional GANs into image description by incorporating target detection features in the input domain. *Tan et al. (2022)* proposed a network composed of two GAN—one mapping the background distribution to produce an image and the other amalgamating description statements. Despite the application of GAN in image descriptions, there remains a need to enhance the semantic richness and accuracy of these approaches. Further research is required to achieve these objectives.

In the domain of visual phrase-guided convolutional neural networks, a notable contribution was made by the authors through the introduction of a phrase-guided message passing structure. This innovative approach facilitates the modeling of dependency information between local visual features by employing an aggregated broadcast message passing mechanism. A relevant work in this context is the fully convolutional scene graph generation model (FCSGG) proposed by *Jin et al. (2023)* which employs relational affinity fields to encode semantic and spatial features in images. FCSGG explicitly represents the relationships between pairs of objects by pointing to integral subregions from subjects to objects, resulting in efficient inference speed. Another significant contribution is

the Visual Translation Embedding (VTransE) model, which explores the modeling of visual relationships by learning mappings of object and relationship features in a low-dimensional space (*Xu et al., 2020*). Building upon VTransE, *Hung, Mallya & Lazebnik (2020)* introduced joint visual translation embedding, which incorporates subject and object features and employs a bidirectional GRU-enhanced semantic embedding model to capture rare relations within a scene. The joint visual translation embedding combines scores from both visual and linguistic modules to rank predictions of triads. In the realm of GANs, several variants have been proposed, including RTT-GAN (*Liang et al., 2017*), LS-GAN (*Qi, 2020*), W-GAN (*Gulrajani et al., 2017*), and CGAN (*Bian et al., 2019*). In this article, we adopt the concept of CGAN to control the generator modality by utilizing target location information in the image, along with target kind information obtained from the scene graph inference. Additionally, to enhance the linguistic richness of the generated image utterances, we incorporate the Senticap corpus for model training (*Mathews, Xie & He, 2016*). Furthermore, to ensure the adequacy of the extracted image features, we integrate an attention mechanism model.

## METHODOLOGY

### Visual relationship fusion model

The article commences by providing a detailed analysis of the bias issue prevalent in existing scene graph generation algorithms. Subsequently, a comprehensive examination of the problem is undertaken, with the structure of this analysis illustrated in Fig. 1.

The model architecture proposed in this chapter focuses on enhancing the common sense-based models, specifically the Structured Neural Model (SNM) and the VTransE model. In the architecture diagram, solid thick arrows represent modules comprised of neural networks, while dashed thick arrows depict the network responsible for inferring the relationship between targets based on fused features and synthesizing target pair labels. Prior to scene graph generation, the chapter initiates a target detection pre-training process. This process utilizes Faster-R-CNN as the underlying target detector. For each image (i), the detector employs a backbone network for extracting image features. Subsequently, an RPN network with inherent anchors is utilized to learn potential target locations and their corresponding object types. Relationship sampling is then performed as part of the process.

The proposed model consists of several key modules. Firstly, there is the target context extraction module, which is responsible for capturing contextual features associated with individual targets. Next, the visual relationship modeling module extracts visual relationship features from subject-object pairs. The label prediction module performs two tasks: predicting the object categories and modeling the relationships, leveraging semantic a priori features. Finally, the relationship context feature extraction module is responsible for extracting relationship context features based on the previously extracted individual object features. The overall algorithm follows the sequence mentioned above, transforming and processing the features accordingly.

Target contextual feature extraction encompasses two main objectives: determining the object label and encoding object features within their context. To achieve effective target

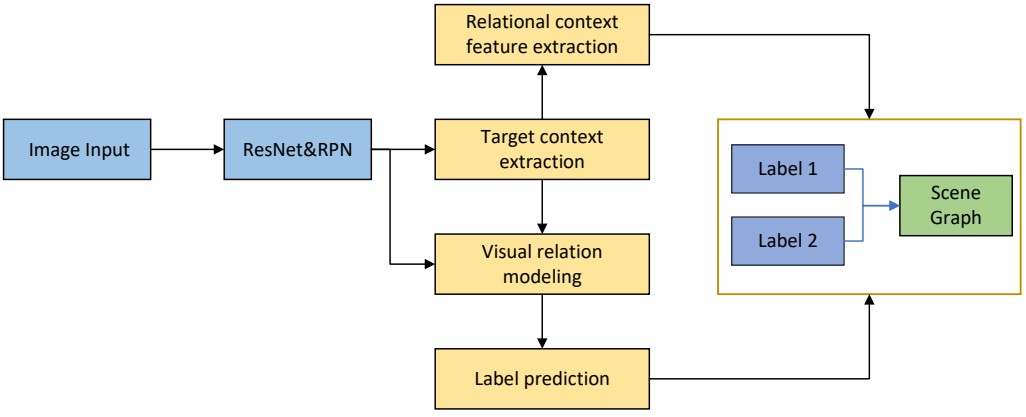

**Figure 1** Structure of visual relation fusion model.

feature extraction and transfer target relationship knowledge, it is essential to refine the training process of target detection by back-propagating the relationship loss to the target detection network. However, the commonly used RoI pooling layer in Faster R-CNN does not employ a differentiable function for coordinate interpolation, as it relies on discrete grid partitioning of proposal regions. This characteristic makes the feature extraction layer susceptible to gradient disappearance. To address this issue, the proposed approach in this chapter replaces the RoI pooling layer with bilinear interpolation, which helps mitigate the problem of gradient vanishing. It is a smoothing function with two inputs, one input is the shape of $W' \times H' \times C$ of the feature mapping M, the other input is the object bounding box projected onto M, the output is the object bounding box of size X\*Y\*C. Each input value can be efficiently interpolated from the map F in a convolutional fashion, as shown in Eq. (1).

$$f_{i,j,c} = \sum_{i'=1}^{W'} \sum_{j'=1}^{H'} M_{i',j',c} k\left(i' - G_{i,j,1}\right) k\left(j' - G_{i,j,2}\right). \tag{1}$$

The visual relation modeling module is primarily utilized to extract image features of objects within an image, specifically the appearance features between pairs of objects that correspond to relations in the image. This module initiates by extracting the proposal regions associated with the target pair and subsequently computes their joint representation. Following this, feature extraction operations such as 2D convolution and batch normalization are applied to the joint representation, which are commonly employed for processing image features. Finally, a pooling operation is conducted in conjunction with bilinear interpolation to facilitate gradient backpropagation. This is illustrated in Eq. (2):

$$v_e = \text{Convs}\left(\text{RoIAlign}\left(M, \text{Convs}\left(b_i \cup b_j\right)\right)\right) \tag{2}$$

where $b_i \cup b_j$ indicates a joint region consists of the proposal area i and the proposal area j.

## Improved GAN networks

Once the scene layout of the image is obtained, the image generation process is performed using the generative network G, which is structured as shown in Fig. 2. The generative network comprises a cascaded refinement network, which is a series of convolutional refinement modules organized in a cascade fashion. Each subsequent layer of the network has twice the spatial resolution of the previous layer, enabling a coarse-to-fine image generation process. The inputs of the model are paired element-wise based on the number of channels, and then a 3x3 convolution operation is applied. The output of each submodule is upsampled using the nearest-neighbor upper-left (UL) operation before being passed to the next submodule. The output from the final module is fed into two convolutional layers to generate the final composite image output.

To better regularize the overall network and improve the learning capability of the model, especially for target samples that occur less frequently in the dataset, this article introduces a GAN module in the network's training process. The GAN consists of a generator that utilizes a cascaded refinement network and a discriminator that employs a fully connected layer. Convolutional neural networks typically have a large number of parameters and deep network layers, which may lack local isovariance properties and suffer from reduced generalization ability with an increasing number of layers. To address this, the article incorporates capsule networks in the discriminator for reconstructing and probabilistically predicting sentences. These reconstructed sentences are expected to be semantically consistent with the images, thereby helping encode the relationship between features. The reconstructed sentences generate reward values under the discriminator and provide feedback to the generator.

The GAN-based regularizer can incorporate target information derived from the target context to ensure that the generated features align well with the target data distribution. This is achieved through a process where the target context is embedded into the GAN framework, influencing both the Generator and the Discriminator. Specifically, target context information is used to:

Guide the Generator: Adjust the Generator's output to better match the target context's characteristics, ensuring that synthetic data or features generated are relevant and realistic.

Refine the Discriminator: Modify the Discriminator's criteria to focus on distinguishing between data points that are not only synthetic or real but also contextually aligned with the target.

## Attention mechanism

Despite the significant advantages of neural networks in image processing, they often lack the ability to differentiate between specific attention and general attention during feature extraction. It is crucial to retain features of certain objects and regions that require focused attention. To address this limitation, this article incorporates the Convolutional Block Attention Module (CBAM), which combines both channel attention and spatial attention mechanisms. This integration allows for better feature representation and preservation of important object and region details.

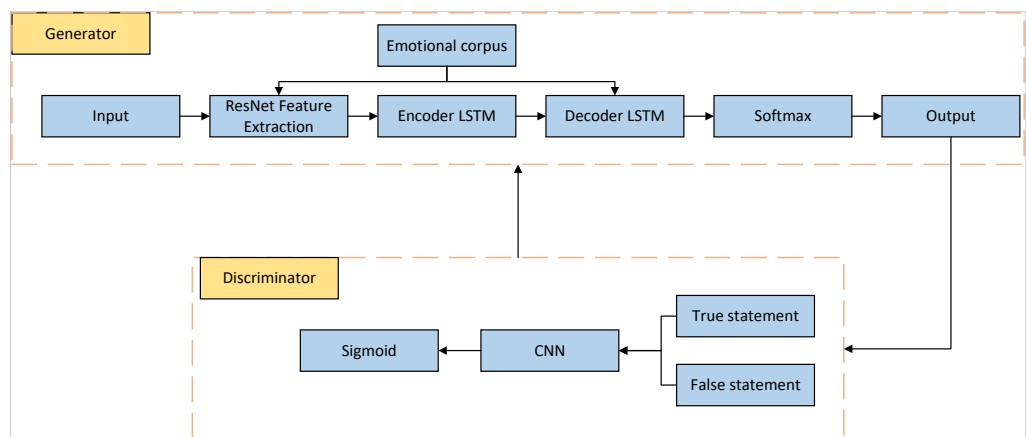

**Figure 2** **Structure diagram based on GAN.**

The scene graph construction module in this article employs two bidirectional long short-term memory (BiLSTM) modules. These modules are utilized to model target contexts and relational contexts, generating refined pairs of target labels and relationships between targets. Specifically, it first organizes the proposal region B into a linear sequence $[(b_1, f_1, l'_1), \ldots, (b_n, f_n, l'_n)]$ , where $f_i$ is the feature extracted for each proposal region $b_i$ is the extracted feature. $l'_1$ is the corresponding label, and then it is fed into the BiLSTM to obtain the object context H , where $H = [h_1, \ldots, h_n]$ , contains the final state of the linearized sequence elements of each proposal region in the BiLSTM, and then each element of the context is decoded according to the previously decoded labels using the LSTM as a decoder to obtain the labels of the target species. $L = [l_1, \ldots, l_n]$ .

For the relationship construction, another BiLSTM is used to propose regions B and object labels L as the input to the relational context C that can be represented as $C = [c_1, \ldots, c_n]$ , which contains the state of the final layer of each proposal region. Finally, combining the target i with the target j together with the contextual features $f_{i,j}$ , the corresponding relational contexts $c_i$ with $c_j$ and the common knowledge information in the dataset to model the final relation $r_{i,j} \in R$ using another LSTM for modeling, the relation R can therefore be constructed in the following way (*Hameed & Garcia-Zapirain, 2020*; *Wei et al., 2022*).

$$H = \text{BiLSTM}\left(\left[f_i; W_1 l_i\right]_{i=1,\ldots,n}\right) \tag{3}$$

$$k_i = \text{LSTM}_i\left(\left[h_i; l_{i-1}\right]_{i=1,\ldots,n}\right) \tag{4}$$

$$l_i = \arg\max(W_o k_i) \in R^{|G|} \tag{5}$$

$$C = \text{BiLSTM}\left(\left[f_i; W_1 l_i\right]_{i=1,\ldots,n}\right) \tag{6}$$

$$c_e = (W_h c_i) \circ (W_t c_j) \circ f_{i,j} \tag{7}$$
$$r_{i,j} = \text{softmax}\left(W_r c_e + W_{o_{i,j}}\right) \tag{8}$$

where $W_i$ is the corresponding mapping matrix for each module, The network architecture incorporates a fully connected layer to implement the sentiment-guided decoding process. During decoding, the input consists of completed sentiment statements, which are learned to guide the LSTM in generating sentiment-infused descriptive statements. In this decoding phase, the output of the encoded LSTM, known as the context vector, is combined with the previously generated words through summation. This process yields the output words for the sentiment description statement.

After obtaining attention weights from both channel and spatial modules, the feature maps are scaled accordingly. This results in more informative features being passed through the network, which can improve the overall performance on tasks such as object detection and classification.

## Loss function

To enhance the optimization process of the generative adversarial network (GAN) for image generation, we propose incorporating perceptual loss and adversarial loss in addition to the conventional pixel reconstruction losses. Traditional GAN training involves alternating between optimizing the discriminator $D$ and the generator $G$, as described by Eqs. (9) and (10).

$$L_D = E_{I \sim p_{\text{rad}}}\left[\log D(I)\right] \tag{9}$$

$$L_G = E_{\hat{i} p_G}[\log(1 - D(\hat{I}))] + \lambda L_i \tag{10}$$

where $\lambda$ is the adjustable parameter, for the loss of the generative network, for better gradient performance, we maximize $\log D\left(G\left(z|S^{\text{layout}}\right)\right)$. For the loss of the reconstructed image comparing the reconstructed image with the original image $L_i$, the distance between the real image $I$ and the synthetic image $\hat{I}$ is calculated, which can be expressed as $L_i = \| I - \hat{I} \|^1$.

To further refine the quality of the generated images, we introduce perceptual loss and adversarial loss into the optimization process. Perceptual loss focuses on capturing high-level features and textures by comparing feature maps extracted from a pretrained network, such as VGG. This loss function operates at the feature level rather than pixel level, ensuring that the generated images preserve high-level details similar to the reference images. The perceptual loss $L_p$ can be formulated as:

$$L_p = \sum_{l \in \text{layers}} \left\| \phi_l(I) - \phi_l(\hat{I}) \right\|_2 \tag{11}$$

where $\phi_l$ represents the feature maps extracted from layer $l$ of the pretrained network.

Adversarial loss, on the other hand, enhances realism by training the generator and discriminator in tandem. The adversarial loss encourages the generator to produce images that are indistinguishable from real images by the discriminator. The total adversarial loss $L_{adv}$ can be expressed as:

$$L_{adv} = \mathbb{E}_{I \sim p_G}[\log D(\hat{I})]. \tag{12}$$

By combining these additinal losses, the updated total loss function for the generator $G$ is:

$$L_G = \mathbb{E}_{I \sim p_G}[\log(1 - D(\hat{I}))] + \lambda_1 L_i + \lambda_2 L_p \tag{13}$$

where $\lambda_1$ and $\lambda_2$ are adjustable parameters that control the influence of the reconstruction loss and perceptual loss, respectively. Incorporating perceptual loss and adversarial loss not only improves the fidelity of the generated images but also enhances their overall visual quality and feature accuracy. This refined approach effectively combines the strengths of pixel-based and feature-based losses, leading to more realistic and high-quality image generation.

# EXPERIMENT AND ANALYSIS

## Formatting of mathematical components

The MSCOCO dataset (DOI: 10.5281/zenodo.7517539) was utilized in the experiment, which includes a minimum of five captions for each image. For the experimental setup, the longest caption in terms of length was selected as the reference. The sentence length was fixed at 40, and any captions shorter than 40 were padded with "<pad>". The training process involved starting the captions with "<start>" and ending them with "<end>". Data augmentation techniques such as "rotation" and "hue transformation" were applied to enhance the dataset. For the augmented dataset, 89% of the images were used for training, while the remaining 11% were used for testing. Additionally, the Senticap corpus, which provides objective and descriptive sentiment-based image utterances derived from MSCOCO, was incorporated. The Senticap corpus contains 1,027 adjective-noun pairs for positive emotions and 436 adjective-noun pairs for negative emotions.

In this article, the evaluation of text sentiment conversion was measured using the accuracy (ACC) metric. The preservation of textual content between the generated text and the original text was assessed using the content preservation rate (CON). The readability of the generated text was evaluated using BELU, where a higher BLEU score indicates better readability and a smaller difference from the original text.

## Results and discussion
### Model comparison

In this article, several models are used for comparison with the proposed approach. These models include:

VCTree (*Tang et al., 2019*): VCTree utilizes a scoring function to compute the task-dependent validity between each pair of objects, forming a tree structure. It employs a bidirectional tree-like LSTM to encode visual relations and a task-specific model for decoding.

IMP (*Xu et al., 2017*): IMP addresses the scene graph inference problem using a standard recurrent neural network (RNN). It improves prediction performance by incorporating contextual message passing.

SNM+TDE (*Tang et al., 2020*): SNM+TDE generates the final inference by subtracting the probability distribution of direct inferences from the scene graph model from the probability distribution of counterfactual inferences from the same model.

VTransE (*Xu et al., 2020*): VTransE is the pioneering approach that applies knowledge representation learning concepts to scene graph generation. It represents relations as embeddings of subject and object differences.

These models serve as baselines to evaluate and compare the performance of the proposed approach in the article. The training process of different models and their accuracy comparison results are presented in Fig. 3. It can be observed that VCTree, IMP, VTransE, and SNM+TDE models reach convergence in approximately 25-30 rounds for VCTree and IMP, and 35-40 rounds for SNM+TDE. In contrast, the proposed model in this article achieves convergence in approximately 10 rounds. Moreover, the accuracy of each data group is significantly improved compared to the other models. This suggests that the proposed model performs better in terms of convergence speed and accuracy on the given dataset.

In addition, performance comparisons of the different models are shown in Figs. 4 and 5.

The proposed model demonstrates superior performance compared to the baseline models in terms of accuracy, BLEU score, and text preservation rate. With an accuracy of 95.42% and a BLEU score of 16.79, the proposed model exhibits enhanced capabilities in visual semantic mining. In contrast, the baseline models achieve high accuracy rates exceeding 85%, but their BLEU scores are significantly lower. This discrepancy can be attributed to training conditions that lack an adequate parallel corpus, causing these models to focus on isolating semantic feature words in the text without considering the coherence of semantic information. Consequently, the generated text from these models outputs sentences with the target sentiment but lacks overall coherence. In contrast, the proposed model employs a self-attentive mechanism to identify sentiment words with extensive sentiment lexicons in the sentences, storing them in the memory network. This approach enables the model to better consider the semantic context, thereby enhancing the text preservation rate and readability of the generated text.

Figure 6 portrays a fortuitous illustration of sentence generation for each model procured through a specific modeling approach. The model introduced in this manuscript showcases a heightened level of semantic cohesion, while exhibiting remarkable similarity to the original text.

The adverse depictions encompass a range of negative human emotions, such as the chilling air, desolate mountain, and solitary street, all characterized by negative adjectives. Conversely, the affirmative portrayals involve pleasant weather, bustling streets, and incorporate neutral terms alongside positive and optimistic adjectives. The VCTree model represents sentence vectors through training that eschews the use of parallel corpora and unrelated content, thereby ensuring the preservation of semantic style within the text. Meanwhile, the SNM+TDE model enhances text fidelity by eliminating specific words that signify textual features, thereby diverting the model's focus from generating sentiment akin to the original text. Furthermore, if the sentiment expressions within the text are cryptic, these models lack the ability to effectively extract the semantic information required to train samples that accurately mirror the original text. The experimental outcomes not only explicate the image content but also manifest emotional nuances, encompassing both

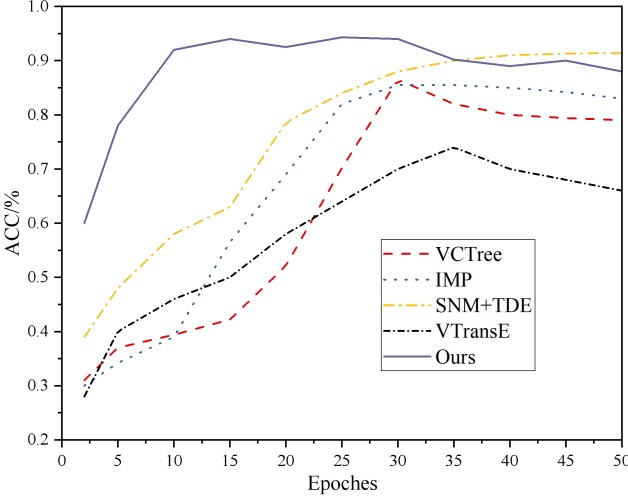

**Figure 3** Accuracy comparison results.

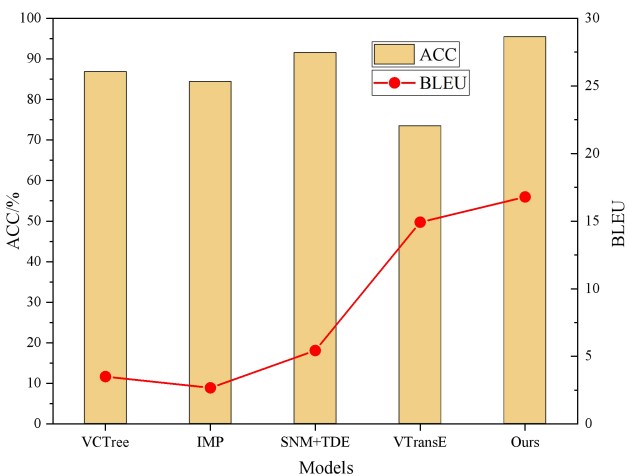

**Figure 4** BLEU and ACC comparison results.

negative and positive human emotions. This serves as compelling evidence of the model's substantial improvement in terms of emotional richness and performance.

## Discussion

Learning with contextual features tends to outperform non-bilinear pooling in various applications. Non-bilinear pooling, which incorporates image features, often negatively impacts the performance of networks such as SNM and VTransE. This is because non-bilinear pooling separates the "style" component of image features from the "content" component of contextual features, which include crucial information for modeling object

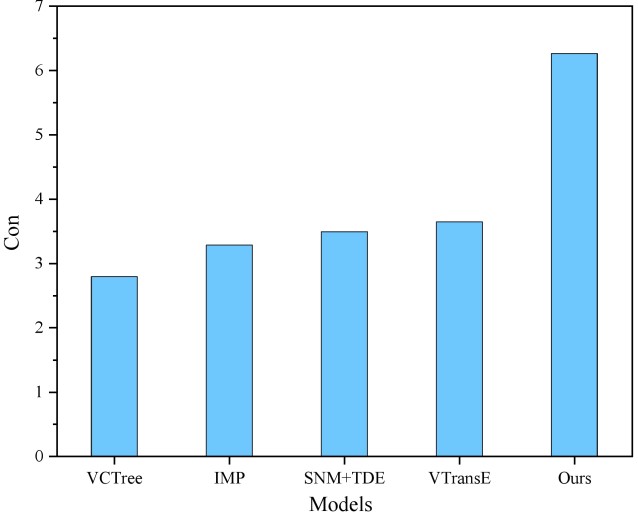

**Figure 5** **Cond comparison results.**

| Input | Moving past the shape, they were dry and truly tasteless. | A man is standing on top of mountain while looking at something far away | This place kicks some serious ass |
|---|---|---|---|
| VCTree | Everyone on the fish, they were fresh and filling | On top of mountain, a man is looking at something far away | Worst service ever. |
| SNM+TDE | Moving past the shape is, they will be wonderful truly. | A man is Looking at the mountains in the distance | It's nice weather here |
| Ours | Moving past the shape, they were tasty and truly delicious | Looking at the cold mountains in the distance, he felt lonely | Very nice here and beat. |

**Figure 6** **Example of emotion generation.**

relationships, such as object labels. The combination of these two components forms the basis for determining object relationships.

It is important to note that the same content can be expressed in different styles. For example, different images depicting a "man near a bird" may have similar contextual features, as they both contain elements of "bird" and "man", indicating a "close" relationship. However, the bird may not always be in a fixed position relative to the person, and it may not always be depicted flying in the sky, resulting in variations in image features. Non-bilinear pooling approaches disregard the correspondence between these factors, leading to reduced effectiveness compared to baseline approaches that solely focus on content.

In the field of art design, accurate generation and expression of emotional semantics can greatly enhance the interactive experience between viewers and artworks. It bridges the gap between the viewer and the artwork, enabling a deeper understanding and engagement with the artistic piece. The innovation of artificial intelligence image creation lies in the ability

to deviate from the original style within a specific range, while adhering to art identification standards and learning from deviations. This involves pushing the boundaries of established artistic styles within a given art category.

By leveraging big data to build relational networks and employing data analysis models and deep learning models developed by data scientists, valuable knowledge can be extracted from existing data. These advanced techniques enable the exploration of new possibilities in art creation and contribute to the evolution and advancement of the field.

## CONCLUSIONS

In this research, we propose a improved GAN model that integrates visual relations with sentiment semantics. The expression of statements conveying positive and negative emotions is accomplished through adversarial training of the generator and discriminator. The generator part of the model extracts image features while incorporating the CBAM attention mechanism, enabling better focus on specific visual regions of interest. On the other hand, the discriminator in this model employs multi-feature map fusion to extract image features, enabling representation of both image details and high-level semantic information. Additionally, a self-coding network is incorporated to downscale the features after the multi-feature map fusion layer, mitigating potential issues related to feature dimensionality. The integration of multi-feature map fusion and GAN modeling in scene graph generation profoundly impacts both image understanding and artistic design. For image understanding, this combined approach achieves a deeper and more accurate interpretation of visual data, which facilitates advancements in applications such as autonomous driving, visual question answering, and content-based image retrieval. However, it is worth noting that this study would benefit from the inclusion of more balanced sampling functions. An intuitive approach to enhance the scene graph generation model involves designing a sampling strategy that corresponds to the long-tail distribution of the entire dataset, as opposed to the current random sampling strategy. Such a strategy would reduce training complexity compared to modifying the model itself.

## ACKNOWLEDGEMENTS

We thank the anonymous reviewers whose comments and suggestions helped to improve the manuscript.

### Funding

The authors received no funding for this work.

### Competing Interests

The authors declare there are no competing interests.

## Author Contributions

- Jiadong Shen conceived and designed the experiments, analyzed the data, performed the computation work, authored or reviewed drafts of the article, and approved the final draft.
- Jian Wang performed the experiments, analyzed the data, performed the computation work, prepared figures and/or tables, and approved the final draft.

## Data Availability

The data is available at Zenodo: Patrick Yu. (2023). MSCOCO tfrecords for meta-dataset [Data set]. Zenodo. https://doi.org/10.5281/zenodo.7517539.

## Supplemental Information

Supplemental information for this article can be found online at http://dx.doi.org/10.7717/peerj-cs.2274#supplemental-information.

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
