# Peer review of "Art design integrating visual relation and affective semantics based on Convolutional Block Attention Mechanism-generative adversarial network model"

_PeerJ Computer Science, doi:10.7717/peerj-cs.2274_

## Round 0.1 · original submission · Major Revisions

Please improve your manuscript thoroughly in light of the reviewers comments and resubmit a detailed response along with the following points

Justify the novelty of the manuscript

Why the proposed GAN model with integrated visual relationships and sentiment semantics surpasses existing approaches, particularly in accuracy and text preservation rate.

Please improve the language quality of the manuscript.

Reviewer 1 ·

Basic reporting

By addressing the following suggestions, your manuscript can effectively highlight the technical innovations, experimental results, and potential impact of your proposed GAN model in scene-based image semantic extraction and artistic design. This will strengthen its contribution to the academic community and enhance its relevance to your intended audience.
The introduction effectively sets up the problem of incongruity between image features and sentiment features due to non-bilinear pooling. To enhance clarity, provide a brief background on why this incongruity is problematic and its implications in practical applications. Expand on the significance of integrating visual relationships with sentiment semantics and how it addresses the identified problem.

Experimental design

Some abbreviations need to be standardized, such as GANs and GAN; VTrans does not provide its full name. Furthermore, abbreviations should only be fully defined upon their first occurrence, but this principle is violated in multiple instances throughout the manuscript.

Consider incorporating perceptual loss and adversarial loss to optimize the current loss function. By comparing feature maps extracted from pretrained convolutional neural networks (e.g., VGG), higher-level image features can be captured, thereby enhancing the quality of generated images.

Validity of the findings

Emphasize how the integration of multi-feature map fusion and GAN modeling contributes to advancing scene graph generation, highlighting its potential impact on image understanding and artistic design.

The manuscript contains technical language suitable for the intended audience. Ensure consistency in terminology and refine sentences for clarity where necessary. Review the manuscript's structure to ensure a logical flow from the introduction of the problem to the presentation of methodology, results, and conclusions.

Additional comments

Why does the introduction to Figure 1 state, 'Figure 1. This is a figure. Schemes follow the same formatting'? Review all errata in the manuscript.
Provide citations for Equations 3-8, as they are not originally authored by the current authors.

·

Basic reporting

The abstract provides a detailed overview of the proposed GAN model and its applications. Consider including a brief statement about the potential impact of improved scene-based image semantic extraction on artistic design.
Remove references to clinical trials and instead emphasize the technical innovations and applications of the proposed GAN model in scene-based image semantic extraction.

Experimental design

The methodology is rich in technical detail, but ensure clarity in how the GAN model integrates visual relationships with sentiment semantics. Provide more specifics on how the GAN-based regularizer operates and its impact on model performance.
Mention the potential impact of improved scene-based image semantic extraction on enhancing artistic design and creative processes.

Validity of the findings

Elaborate on the training process, including the specific role of the GAN-based regularizer and how it incorporates target information derived from the target context. Clarify the function and implementation of the Convolutional Block Attention Mechanism Model (CBAM) and its impact on feature extraction.

Additional comments

Discuss in detail why the proposed GAN model with integrated visual relationships and sentiment semantics surpasses existing approaches, particularly in accuracy and text preservation rate.

---

## Round 0.2 · accepted · Accept

Hi, Based on the input received from the experts on your revised version. I am pleased to inform you that the experts are now satisfied and recommending your article for publication.

Reviewer 1 ·

Basic reporting

Manuscript has been revised as per given directions

Experimental design

Ok

Validity of the findings

NA

Additional comments

NA

·

Basic reporting

'no comment'

Experimental design

'no comment'

Validity of the findings

'no comment'

Additional comments

'no comment'